# A case study of transferring the effect of demographic factors on e-waste recycling to the waste container assignment model

Tuba Bazancir[1], Yeşim Ok[2]*

1 Quality Coordination Department, Trabzon University, Trabzon, Turkey, 2 Industrial Engineering Department, Ataturk University, Erzurum, Turkey

* yesim.ok@atauni.edu.tr

## Abstract

Collecting waste directly from the consumer could be considered the initial stage of electronic waste recycling. Despite numerous studies employing mathematical assignment models to determine the optimal location of containers for waste collection from end-users, no research has been deemed to incorporate the demographic characteristics of the regions into these assignment models. This study aims to accomplish more effective results by incorporating the factors affecting consumers' recycling trends (the population, income level, education level, and age distribution of the relevant region) into the assignment model. By integrating abovementioned parameters characterizing regional differences in the micro sense and country-based differences in the macro sense into the assignment model, a generic model is structured to help local administrators determine their first-step e-waste recycling policies. The main contribution of the proposed model is that significantly more realistic results regarding the amount of e-waste to be collected can be obtained when factors indicating regional differences are included in the container assignment model. Thus, local governments could utilize these findings as a reference to create more sustainable policies for the next steps of e-waste recycling. The model application is demonstrated through a case study for a local government in Turkey.

## Introduction

Electronic waste (e-waste) is one of the world's fastest-growing sources of waste [1,2]. While the variety and quantity of electronic items grow in tandem with rapid changes in consumer behavior, their usable life reduces with every passing day. According to the Global E-waste Monitor Report 2020, 53.6 million tons of electronic waste were generated in 2019. It is anticipated that worldwide electronic waste will exceed 74 million tons by 2030. This number is nearly double the amount of e-waste

**Data availability statement:** All relevant data are within the manuscript and its Supporting Information files.

**Funding:** The author(s) received no specific funding for this work.

**Competing interests:** The authors have declared that no competing interests exist.

generated in 2014. Furthermore, only 17.4% of the e-waste generated in 2019 was recycled officially [3].

The difficulty of managing waste electrical and electronic equipment has started to damage electronics' position as a "clean" technology, and the issue has grown in relevance as the electronics industry expands at one of the most rapid rates among all economic sectors [4].

E-waste left to nature could take several years to disintegrate, and the hazardous substances in its contents interacting with soil and water can cause a variety of detrimental effects. On the other hand, appropriately processed waste can provide a secondary source of raw materials for industry. Therefore, efficient management of electronic waste is vital for sustainability in many aspects. Even in developed countries, this recycling issue has not been completely and successfully addressed. For emerging countries like Turkey, it is a complex equation with many unknowns that require solutions. While it is acknowledged that manufacturers of electronic waste are involved in the process of implementing laws in developed economies, policymakers in developing countries frequently overlook the root causes because they place too much emphasis on the financial aspect of e-waste [5].

One of these sub-problems is allocating e-waste containers to places that will permit access to the most e-waste. Local governments often do not regard waste container assignment to be a serious issue, and therefore randomly allocate the same number of waste bins to all regions within their jurisdiction. As a result, they incur additional expenditures and produce inefficient results by failing to differentiate between locations with low and high e-waste generation potential. However, the question that must be addressed is: why does the amount of waste differ by region? Several crucial factors influence waste generation and container placement. A considerable number of research on recycling behavior indicate that socio-demographic characteristics significantly influence recycling habits [6–9], while others claim that these factors play only a minimal role [10].

The decision-makers in municipal solid waste management systems have to consider a number of processes, including waste collection, transportation, treatment, and disposal. Uncertainties also surround a large number of system characteristics, influencing factors, and their interconnections. These uncertainties may be further increased by temporal and spatial variations in numerous system components [11].

Sociodemographic characteristics encompass a range of social and demographic factors, which are often measured by an individual's age, level of education, occupation, and income [12]. The main purpose of this study is to create a generic assignment model based on the sociodemographic characteristics that influence e-waste generation and container placement.

While survey-based studies have identified the qualitative and quantitative factors influencing e-waste recycling behavior, a significant gap remains in establishing a framework that incorporates these factors into the planning of the e-waste recycling process. This study seeks to address this gap, specifically by incorporating quantifiable socio-demographic factors into the waste container allocation model, addressing these factors at both macro and micro levels.

Thus, local governments that use this generic model will be able to develop a specific application by incorporating their conditions into the model, meeting municipalities' demand for methodical and scientific studies to guide the e-waste collection process.

## Literature

Some research on e-waste recycling has typically employed multi-purpose models to address both environmental and economic concerns [13]. Among these studies, [2] suggested a DEMATEL method that helps lower transportation costs, emissions, and fossil fuel use, while [14] applied a fuzzy DEMATEL approach to create an e-waste collection policy in India, taking into account important factors like technology involvement and environmental programs.

Another issue that is under consideration is vehicle routing for e-waste pickup. A study conducted in southern Poland [15] proposed a model that utilizes a genetic algorithm to optimize daily schedules and routes for vehicles collecting mobile electronic waste. [16] suggested two models, one focused on the flow of goods and the other on specific locations, to solve the problem of planning where to place collection points and how to route vehicles for a company that collects used products from customers in exchange for financial rewards. [17] developed a mathematical model to identify the best locations for collecting e-waste and the best routes for vehicles based on opportunity costs.

Research that considers the affect of consumer attitudes on e-waste recycling has additionally received attention. Consumer participation has been considered one of the most critical factors contributing to the effectiveness of e-waste recycling management [18]. Therefore, it is crucial to understand consumers' disposal and recycling behavior to increase residential recycling rates. Survey-based research makes up the majority of the literature on consumers' e-waste recycling behavior [19,20]. Since the usage of electronic devices is most commonly associated with young people, surveys of university students come to the forefront [21,22]. These surveys also have made it simpler to acquire data on the lifespan of electronic equipment [23,24]. Consequently, demographic profile data, such as gender, age, educational status, and monthly income, as well as subjective factors, such as awareness, responsibility, perceived benefits, and risks, could be used as input data to determine the consumers' intentions of recycling e-waste [25].

The current literature uses various deterministic models for optimization of the different steps of the e-waste recycling process. [26] proposed a reverse logistics network model that aims to reduce total cost, considering the cost of collection, the cost of installing separation, repair, and recycling facilities, processing capacity, and transportation between e-waste facilities. For a more realistic LP model under uncertainty, [27] defined parameters such as demand (the amount of e-waste to be processed), the operating capacity of resources, the quantity of e-waste output, and processing times.

[28] developed a mixed integer linear programming (MILP) model that allows a single waste collector to be allocated to several communities within a neighborhood, considering both the total waste separation volume of collectors and their proximity to the neighborhood. [29] has proposed a two-objective mixed integer nonlinear model for scenarios where consumers deposit their e-waste at a collection center and receive a fixed financial incentive or where e-waste is collected at their doorstep without receiving any financial incentive. The objectives are to minimize the collection costs and maximize the amount of WEEE collected. For the WEEE drop-off case, two factors, namely incentive and ease of disposal (i.e., distance between the customer node and the collection center), are considered.

In the MILP model constructed in [30] the optimum number and location of collection points, collection centers, and recycling centers were determined to minimize the overall costs, which include the fixed costs associated with e-waste facilities and the transportation costs between these facilities. The study focused on minimizing e-waste costs [31], including collection, transportation, and processing, while aiming to maximize resource recovery and reduce environmental impact through a multi-objective MILP model. However, none of the above models has accounted for sociodemographic factors affecting e-waste demand in the assignment model.

A binary logistic regression analysis was done using survey results to see how factors like income, age, education, knowledge, habits, traditions, incentive payments, and behaviors affect people's willingness to recycle mobile phones [32].

Furthermore, the literature includes studies that incorporate financial incentives into the assignment model as a factor influencing e-waste consumption [29,33]. However, no research has identified sociodemographic factors affecting e-waste demand as a direct component in the container allocation model.

In the current literature, there is a lack of research that directly integrates socio-demographic characteristics affecting e-waste consumption within the container allocation model. The main motivation of this study is to create a general framework that would aid in the development of local e-waste collection policies based on the basic socio-demographic factors that cause variations from region to region in the e-waste recycling rate.

## Materials and methods

### Classification of electronic waste

Electronic items turn into electronic waste when they complete their useful lives. Electronic waste is formed as follows: first, electronic goods are sold. Then these items are used in homes and workplaces. When they complete their useful lives, electronic waste is formed. Finally, they take part in the collection and recycling phases. E-waste encompasses a wide range of products.

The European Union (EU) Waste Electrical and Electronic Equipment (WEEE) Directive divides e-waste into six categories (as shown in Fig 1) based on recycling and recovery targets [34].

Large household appliances, computers, cell phones, televisions, medical equipment, sports gear, toys, and other discarded electronic devices are all considered e-waste. However, the model in this study includes e-waste that the consumer transports to the collection points on their own as categories 2 and 6, colored red in Fig 1. Although the majority of research in the e-waste literature used the amount of e-waste as a parameter in their models, in this study, it is derived from the model solution.

### Factors affecting e-waste recycling

Recycling necessitates individual effort due to the need to separate, prepare, and store e-waste. So, the decision to recycle is complex and involves various factors. When some of the studies are examined regarding the willingness for e-waste

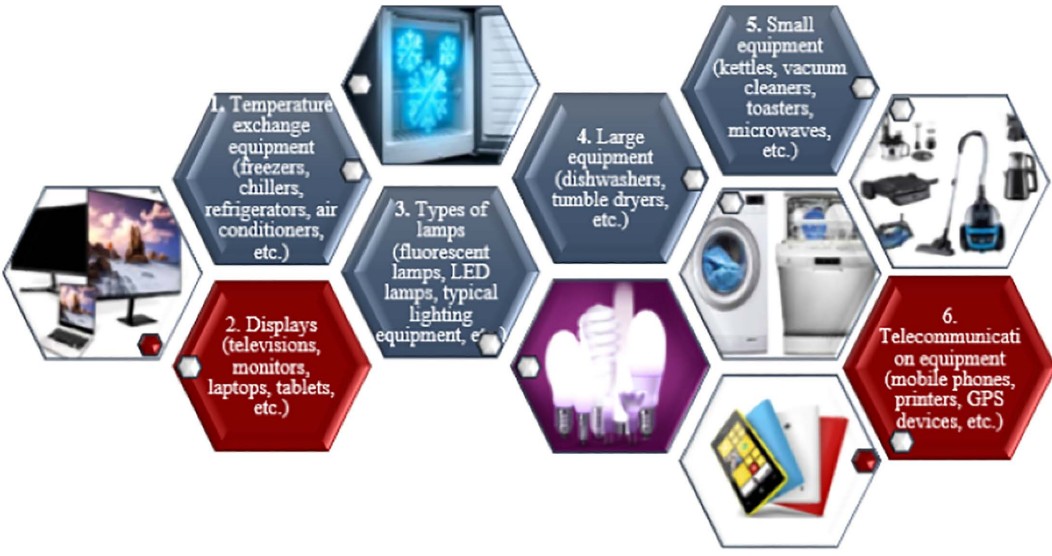

**Fig 1. WEEE categories.**

recycling; providing ease of collection and receiving some financial incentives for product returns work well [35] gender, age, and environmental awareness are important determinants [6], as are age, general awareness and general attitude towards the environment [36].

According to a survey in the USA [10], the strongest determinants of the intention to recycle e-waste are personal and moral norms, environmental awareness, and social expectations, while gender, marital status, and suitability of recycling have a smaller effect. Through hypothesis testing using survey data, a WEEE recycling behavior model for Romanians was developed, and it was found that people's attitudes and habits were the main determinants of their e-waste recycling behavior [37]. Using the logistic regression model, it is also found that education level, age, and household income are important and dominant factors in WEEE behavior [38].

The Theory of Planned Behavior (TPB) is one of the most popular models preferred in behavioral research on recycling and has been used in many studies on the determinants of e-waste recycling [22,39–42]. In its simplest form, TPB explains behavioral intention as a function of three components: attitude, subjective norm, and perceived behavioral control. In their highly cited study, [40] included the three main components of TPB, as well as socio-demographic factors, the degree of awareness of the problem, and the environmental situation of Brazil. Previous studies have shown that, apart from the behavioral aspects of consumers' e-waste recycling behavior, there is also a relationship between demographic and socioeconomic variables such as household gender, age, education, and income [8,6].

Different from the existing literature, the proposed model integrates factors affecting e-waste recycling into the container allocation model by considering them in both macro and micro dimensions, as illustrated in Fig 2. The reason for preferring these factors for e-waste recycling in this study is their ability to be included in the mathematical model with quantitative values.

## Model assumptions

Various scenarios concerning the collection and transportation of electronic waste may require optimization. The location of collection sites and treatment facilities is one of the most frequently addressed subjects among them. One aspect that influences consumers' propensity to recycle is the collection point's proximity to demand [16]. Unlike earlier research

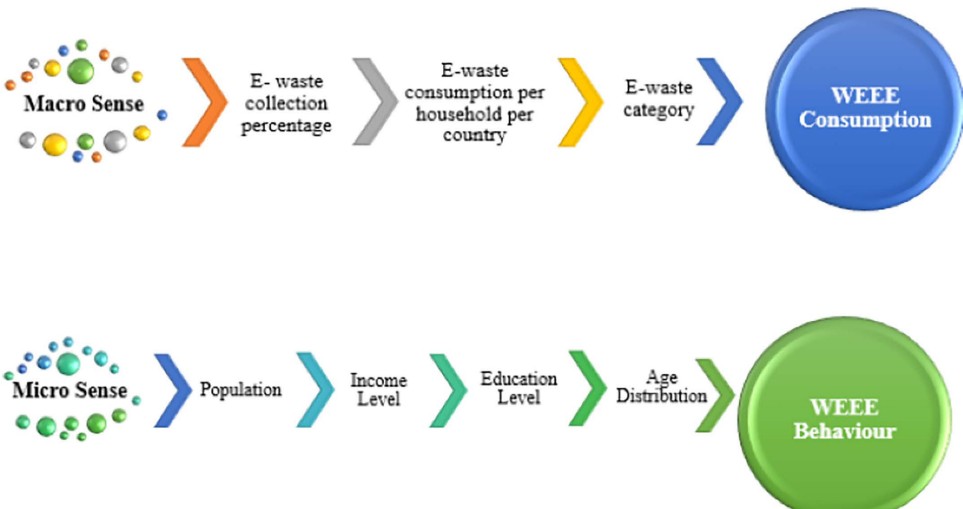

**Fig 2. Country-based differences affecting WEEE consumption in the macro sense and Socio-economic/demographic factors affecting WEEE behavior in the micro sense.**

[43,44], the model proposed in this study incorporates the factors affecting electronic waste recycling into the assignment model as well.

The assumptions of this study could be listed as follows:

- The model incorporates a capacity restriction to represent its assumption that every neighborhood generates e-waste demand based on its unique socio-demographic characteristics.

- Due to the abovementioned factors being incorporated as parameters in the container assignment model, the e-waste amount is predictable approximately.

- As the harmonic mean eliminates the extreme points and reduces the effect of averaging the high value between the data, it is preferred in this study for factors combination.

- Even though the application is intended for a local government region in Turkey, as the model has a generic structure because of its general parameters, local policymakers in any nation could use it to gain insight while determining their initial e-waste policies.

## Model formulation

We define the identity set of all demand nodes (DN) as I= {i| i = 1, 2, 3… m} and the identity set of all candidate container points (CP) as J= {j| j = 1, 2, 3…n}.

$DC$ is the maximum coverage distance between each DN and its corresponding candidate CP. Therefore, for each DN, the identity set of candidate CPs to which a DN can be assigned is calculated as $N_i = [j \mid d_{ij} \leq Dc, \forall j \in J] \forall i \in I$, where $d_{ij}$ is the distance between $i^{th}$ DN to the $j^{th}$ CP.

Also, the following parameters are defined for model formulation:

CN(i) is the number of containers in $i^{th}$ DN, where $CN_{max}$ is the maximum number of containers that can be placed with the budget available. Here, the budget constraint allocated by municipalities for e-waste recycling is considered.

Since e-waste recycling rates vary from country to country, two more parameters were added to the model formulation. $E_{wcr}$ is the e-waste collection percentage by country and Ewcp is the e-waste amount per household by country (per house).

Except for these parameters, HN (i) is the number of houses in $i^{th}$ DN and S is the capacity of each container, as the parameter in which socio-demographic factors of the relevant neighborhood are included in the model. HMF (i) is the district-based harmonic mean of socio-demographic factors affecting e-waste recycling.

The decision variables in the model are:

$$x_{ij} = \begin{cases} 1, & \textit{if container j is placed in district i} \\ 0, & \textit{otherwise} \end{cases}$$

$$y_{ij} = \textit{the amount of e − waste thrown into the container j from district i}$$

The objective function is:

$$\text{Max} \sum_{i \in I} \sum_{j \in J} y_{ij} \tag{1}$$

The constraints are:

$$\sum_{j \in Ni} x_{ij} \geq 1, \qquad \forall i \in I \tag{2}$$

$$\sum_{j \in Ni} x_{ij} \leq CN(i), \qquad \forall i \in I \tag{3}$$

$$\sum_{i \in I} \sum_{j \in Ni} x_{ij} \leq CN_{\max} \tag{4}$$

$$\sum_{j \in Ni} y_{ij} \leq HN(i) * Ewcr * Ewcp, \forall i \in I \tag{5}$$

$$y_{ij} \leq HMF(i) * S * x_{ij}, \forall i \in I, \forall j \in Ni \tag{6}$$

$$HMF(i) = i/(1/t_i), \qquad \forall i \in I \tag{7}$$

$$x_{ij} \in \{0, 1\} \quad \text{and} \quad y_{ij} \geq 0 \quad \forall i \in I, \forall j \in J \tag{8}$$

Equation 1 aims to maximize the amount of e-waste to be collected. Equation 2 ensures that e-waste containers are assigned to every neighborhood within the coverage area. This uses the previously described formulation of the set Ni. Candidate container allocations are limited to regions within the maximum coverage distance, similar to the model used by [45] for the distributed energy network design. Equation 3 limits the number of containers for each region according to the general situation. Equation 4 is a budget constraint. The maximum number of containers that can be used in the model is obtained by dividing the budget of the relevant local administration by the container unit cost. Equation 5 limits the maximum quantity of e-waste for a region. Since the model is generic, this value can be calculated by multiplying the number of houses in the relevant region by the amount of e-waste per household by country and the e-waste collection percentage for each country according to the global e-waste statistics. Equation 6 is the capacity constraint and the harmonic mean value of factors (HMF) in Equation 6 is derived from Equation 7. The harmonic mean values for each neighborhood are multiplied by container capacity, resulting in an impact restricting the amount of e-waste in the model. Equation 7 includes the harmonic mean formula. The values $t_1$, $t_2$, $t_3$, and $t_4$ reflect the population, income, education, and age distribution ratios, respectively. Finally, Equation 8 defines the decision variable x as 0–1, and y as a continuous variable.

The model comes to the fore with a different perspective proposal as it creates an e-waste demand based on the different characteristics of each neighborhood. The harmonic mean is preferred in this model because it eliminates the extreme points and reduces the effect of averaging the high value between the data.

## Case study

### The general flow of the research methodology

Fig 3 demonstrates the flow of the novel approach that has been suggested, in which the parameters characterizing local differences in the micro sense and country-based differences in the macro sense are both integrated into the assignment model.

### Constitution of parameters

**Number of houses in selected neighborhood (HN (i)).** The number of houses belonging to the selected Neighborhood has been given (in Table 1, in the appendix).

**Distances between demand points and candidate container points ($d_{ij}$).** Euclidean distances are calculated using random coordinates generated at specific intervals for the demand and candidate container points that present the Euclidean distances, $d_{ij}$, between demand point $i$ and candidate container $j$ in meter scale (Table in S2 Table).

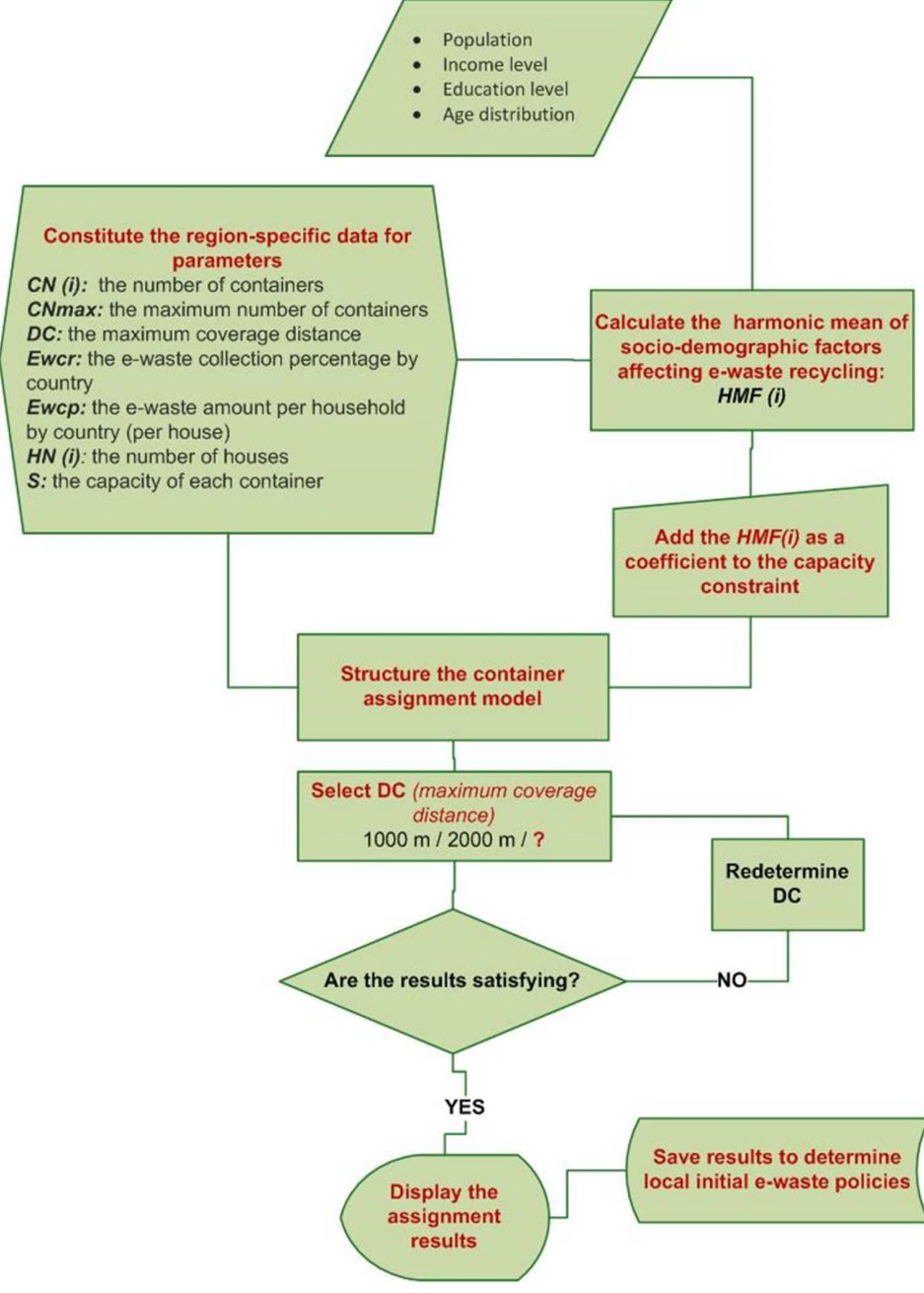

**Fig 3. The general flow of the research methodology.**

**E-waste amount per household ($E_{wcp}$).** E-waste encompasses an extensive variety of products. As previously stated, e-waste is categorized into six classes in accordance with the European Union WEEE Directive (Fig 1). This study focuses on the collection of e-waste categories ranked second (displays) and sixth (telecommunication equipment). The application is restricted to these two categories because items like monitors, tablets, mobile phones, and headphones are of sizes and weights portable by household and the container capacities designated in the model are appropriate for

**Table 1. Objective function values for models include and exclude factors in different coverage distances.**

| | Objective Function Value (Amount of e-waste collected) (kg) | | | | | |
|---|---|---|---|---|---|---|
| Coverage Distance – Dc (m) | *500* | *750* | *1000* | *1500* | *2000* | *2500* |
| Factors Included | 1396.5 | 2463 | 2956.5 | 3570 | 3972 | 3972 |
| Factors Not Included | 13500 | 22500 | 27000 | 31500 | 34500 | 34500 |

this context. For instance, items like refrigerators in category 1 and washing machines in category 4 are not suitable for households to transport to the nearest disposal container. Once more, light bulbs in category 3 were excluded from the model due to their hazardous contents that could be released upon breakage. Category 5 was also excluded from this model due to its composition mostly consisting of waste that is likely to surpass the container's capacity.

According to the data in the Beyond Waste report, the total amount of e-waste in Turkey for 2016 is 542 thousand tons, while the amount of e-waste in the televisions/monitors and informatics-telecommunication equipment categories of e-waste categories is 117 thousand tons [46]. Since only the two e-waste categories are taken into consideration in this study, an approximate percentage is obtained by dividing the e-waste amount in these two categories by the total e-waste amount (calculated as 117/542 = 0.215).

According to the research conducted by the "Clear it waste collection" company, e-waste amounts per household have been determined for some European countries [47]. According to this research, the amount of e-waste per household in Turkey is 41.8 kg/year.

The anticipated amount of e-waste categories of televisions/monitors and informatics-telecommunication equipment per household could be calculated by multiplying the amount of e-waste per household in the relevant country by the rate of e-waste in selected WEEE categories. For this study this value is approximately 9 kg/year. This value is consistent with the values obtained in Salihoğlu and Kahraman' s study [48] evaluated the e-waste consumption in Bursa, Turkey through a survey.

So, the e-waste amount per household by country $(E_{wcp})$ is calculated for Turkey as 9 kilograms. The monthly value is calculated by dividing the annual value per household by 12 in Turkey. Here, the 0.75 value obtained for e-waste categories of televisions/monitors and informatics-telecommunication equipment, will be multiplied by the number of houses in each neighborhood, in the constraint in Eq. 5 in the model.

**E-waste collection rate by country $(E_{wcr})$.** The e-waste collection rate varies country by country. The E-waste collection percentage for Turkey $(E_{wcr})$ was 0.18 in 2019 [49].

**Capacity of an e-waste container $(S)$.** S is the container capacity; the capacity of a container is taken as 4.5 m³/1500 kg. A larger capacity is not needed as it is for only televisions/monitors and informatics-telecommunication equipment. However, larger or smaller-sized e-waste containers could be included in the process by changing the right-hand side value in the capacity constraint (Eq. 6) in the model.

**Maximum number of containers $(CN_{max})$.** The budget allocated by the municipality for e-waste containers is taken as 300.000 ₺. The cost of the unit e-waste container is determined as 13.000 ₺ in line with the information obtained from authorized institutions. The municipality's budget is divided by the unit box cost and the maximum number of containers that can be used in the selected neighborhoods is calculated as in Eq. 9.

$$CN_{max} = \text{Total budget of Municipality} / \text{Unit box cost} \qquad (9)$$

**Candidate container numbers for each neighborhood $CN\ (i)$.** The potential number of e-waste containers is calculated by dividing the total budget by the unit container price, utilizing the budget allocated by the local government for recycling activities. The maximum number of containers defined in the manuscript is 23. Fig 4 illustrates the distribution

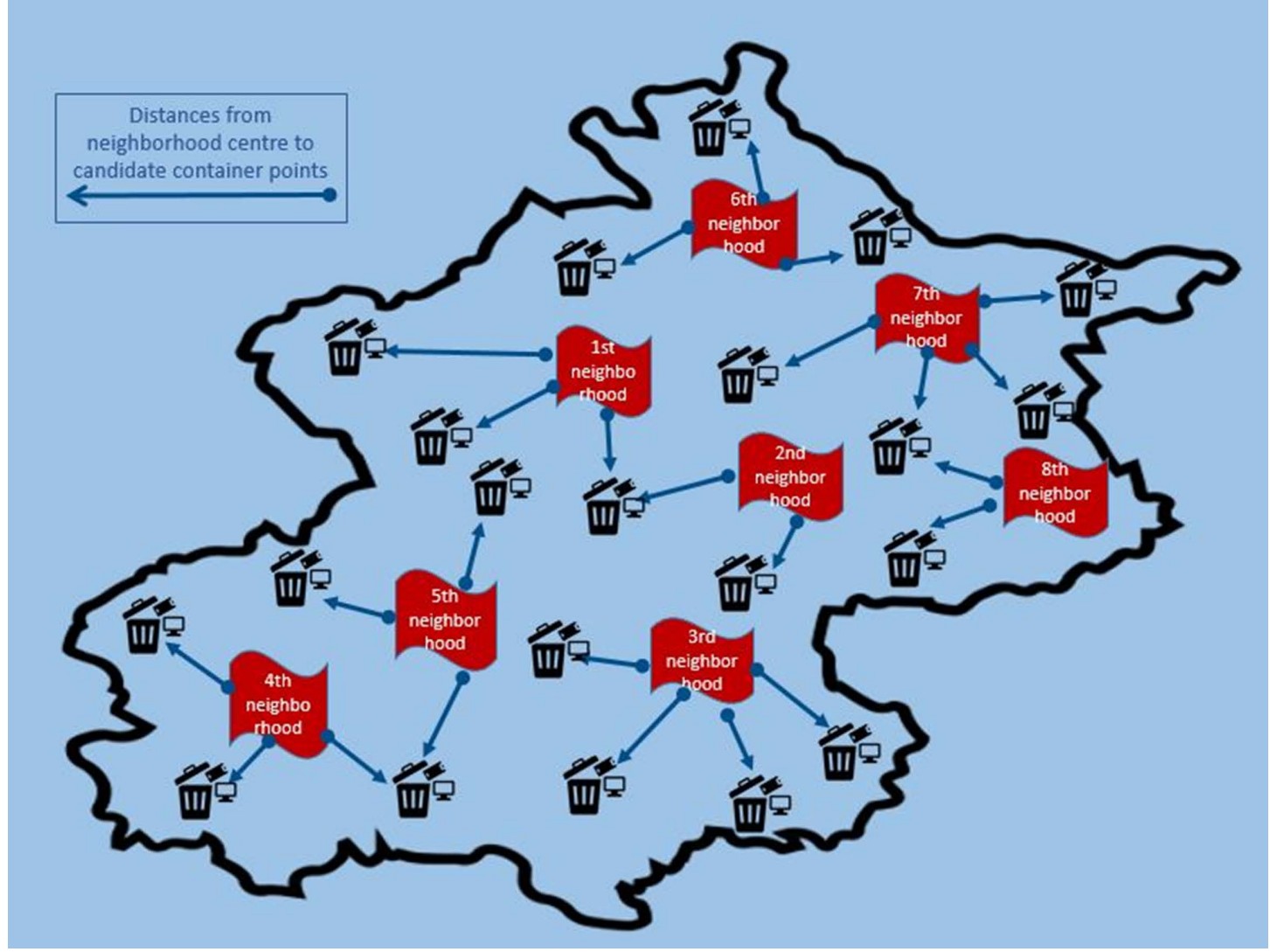

**Fig 4. Candidate container and demand points relationship illustration.**

of candidate container points across eight neighborhoods. The number of containers assigned to each neighborhood is detailed in Table in S3 Table.

**Calculation of harmonic mean values.** Per capita income and population information for all neighborhoods of Yakutiye district were obtained from the Index Platform [50] and these values are listed (Table in S4 Table).

It has been observed that the use of electronic goods is associated with the level of education. According to the results of [23]; it is concluded that the age range of 18–24 is the age range in which most electronic devices are used, therefore the potential to create televisions/monitors and informatics-telecommunication equipment categories of e-waste is much higher in the young age group than the middle and advanced. In this context, in our study, a scale between 1 and 10 is used for scoring the level of education.

Neighborhoods where the 20–29 age range is predominant are given 10 points, and 7 points for the 0–19 age range. The education level scores, education scoring, and age distribution scoring have been determined (Tables in S5 and S6 Tables).

The population ratio for all neighborhoods is normalized by dividing the population of each neighborhood by the total population. The same normalization process is performed for per capita income, education, and age distribution scores, and the relevant ratios are calculated (Table in S7 Table).

After the rate calculation of the four factors for all neighborhoods, these ratios have been reversed in turn, calculating the equal weight harmonic means for each one. For example, for the 1$^{st}$ Region (Shukrupasa Neighborhood), the equally weighted harmonic mean value is calculated as in Equation 10.

$$HMF(1) = \frac{4}{\left( \frac{1}{0,216} + \frac{1}{0,114} + \frac{1}{0,024} + \frac{1}{0,10} \right)} = 0,061$$

(10)

According to the calculated harmonic means, the neighborhoods have been sorted (Table in S8 Table), starting with the larger one.

The harmonic mean values of the four factors that are frequently underlined in the literature regarding e-waste consumption are calculated with the formula in Equation 7 and used in the mathematical model. The reason for choosing the harmonic mean is to eliminate the extreme values.

## Results

### Scenario analysis

The results of the proposed model were evaluated with different scenario analyses.

In this context, the effect of adding macro and micro parameters to the assignment model for e-waste container assignment on the result obtained is given in Tables 1 and 2 at different coverage distance sizes. When the results in both tables are examined, the results can be evaluated in two dimensions.

The values labeled as "factors not included" in Table 1 represent the values that would be obtained if the proposed model was run without considering the demographic factors that affect the amount of e-waste consumption; they reflect the effect of the factors on the model.

Initially, the parameters added to the proposed model have a greater impact on the objective function: the amount of e-waste expected to be collected so much so that the expected amount of waste could be almost ten times higher when the factors are excluded. So, this proposed approach, deemed effective in the initial phase of e-waste recycling decisions by local authorities, can prevent the allocation of unnecessary containers and associated costs.

Secondly, as the coverage distance increases, the number of possible containers (that is, the number of assignments) in which each neighborhood can throw its waste increases, but it is seen that the number of containers assigned decreases relatively. When the coverage distance is changed on the scale of 500–2500 meters, the number of containers placed by the model varies between 10 and 16. The number of zones assigned to containers increases as the coverage distance increases. What is meant by increasing the coverage distance here is to take into account the possibility of consumers living

**Table 2. Results for models including and excluding factors in different coverage distances.**

| Number of containers located | | | | | | | Number of assignments | | | | | |
|---|---|---|---|---|---|---|---|---|---|---|---|---|
| *Coverage Distance – Dc (m)* | 500 | 750 | 1000 | 1500 | 2000 | 2500 | 500 | 750 | 1000 | 1500 | 2000 | 2500 |
| *Factors Included* | 10 | 16 | 16 | 15 | 15 | 15 | 13 | 17 | 18 | 21 | 23 | 23 |
| *Factors Not Included* | 10 | 16 | 16 | 13 | 14 | 16 | 13 | 17 | 18 | 21 | 23 | 23 |

in a neighborhood throwing their waste into containers further away from them. However, in actuality, it is more likely that as it becomes more difficult for end users to access e-waste recycling containers, their willingness to recycle decreases.

By extending the scenario analysis, the container assignment results of the parameters added to the model for Dc = 1000 m and Dc = 2000 m are given in detail in Tables 3(a) and (b).

The objective function value of the model with 1000 m Dc value (2956.5 kg) is lower than the objective function value of the model with 2000 m Dc (3972 kg). Therefore, when the maximum coverage distance is reduced, it is seen that the number of containers increases, but the expected amount of e-waste decreases, too.

The assignments changing depending on the maximum coverage distance determined in the model are visualized in Figs 5 and 6. Comparing above mentioned figures, it is obvious that as the coverage distance between the end user and the e-waste containers increases, fewer containers are allocated (16 containers in Fig 5 and 15 containers in Fig 6).

This is because, in regions where neighborhood boundaries are close together, multiple neighborhoods can use the same container as is already the practice case. Fig 5 illustrates this scenario with the assignment of three different regions (4, 5, and 8) to container number 18.

Because of this, the coverage distance should be determined by considering the necessity of not taking the maximum coverage distance too high and placing e-waste recycling bins at points where the end user can reach them as easily as possible. As a result, with this model, it is recommended to determine the most appropriate distance with various trials instead of selecting the coverage distance as a large or small value.

Table 3. (a) Assignment results for maximum coverage distance of 1000 m (b) Assignment results for maximum coverage distance of 2000 m.

| Z = 2956.5 (a) Dc = 1000 m | | | | | | | | | Z = 3972 (b) Dc = 2000 m | | | | | | | | |
|---|---|---|---|---|---|---|---|---|---|---|---|---|---|---|---|---|---|
| Candidate Points | 1 | 2 | 3 | 4 | 5 | 6 | 7 | 8 | Candidate Points | 1 | 2 | 3 | 4 | 5 | 6 | 7 | 8 |
| 1 | 1 | – | – | – | – | – | – | – | 1 | 1 | – | – | – | – | – | – | – |
| 2 | – | – | – | – | – | – | – | – | 2 | – | 1 | – | – | 1 | – | – | – |
| 3 | – | – | – | – | – | – | – | – | 3 | 1 | – | – | – | – | – | – | – |
| 4 | – | – | – | – | 1 | – | – | – | 4 | 1 | 1 | – | 1 | 1 | 1 | – | – |
| 5 | – | 1 | – | – | – | – | – | – | 5 | – | – | – | – | – | – | – | – |
| 6 | – | 1 | – | – | – | – | – | – | 6 | – | – | – | – | – | – | – | – |
| 7 | – | 1 | – | – | – | – | – | – | 7 | – | – | – | – | – | 1 | – | – |
| 8 | – | 1 | – | – | – | – | – | – | 8 | – | – | – | – | – | – | 1 | – |
| 9 | – | – | 1 | – | – | – | – | – | 9 | – | – | 1 | – | – | – | – | – |
| 10 | – | – | – | – | – | – | – | – | 10 | 1 | – | 1 | – | – | – | – | – |
| 11 | – | – | – | – | – | – | – | – | 11 | – | – | 1 | – | 1 | – | – | – |
| 12 | – | – | – | – | 1 | – | – | – | 12 | – | – | – | 1 | – | – | – | – |
| 13 | – | – | – | 1 | – | – | – | – | 13 | – | – | – | 1 | – | – | – | – |
| 14 | – | – | – | 1 | – | 1 | – | – | 14 | – | – | – | – | – | – | – | – |
| 15 | – | – | – | – | – | – | – | – | 15 | – | 1 | – | – | – | – | – | – |
| 16 | – | – | – | 1 | – | – | – | – | 16 | – | – | – | – | – | – | – | – |
| 17 | – | – | – | – | – | – | – | – | 17 | – | – | – | – | – | – | – | – |
| 18 | – | – | – | – | – | – | – | – | 18 | – | – | – | – | – | – | 1 | 1 |
| 19 | – | – | – | – | – | 1 | – | – | 19 | – | 1 | – | – | – | – | – | – |
| 20 | – | – | – | – | – | – | 1 | – | 20 | – | – | – | – | – | – | – | 1 |
| 21 | – | – | – | – | 1 | – | – | – | 21 | – | – | – | – | – | – | – | – |
| 22 | – | – | – | – | – | – | 1 | 1 | 22 | – | – | – | – | – | – | – | – |
| 23 | – | – | – | – | – | – | – | 1 | 23 | – | – | – | – | – | – | – | – |
| CN | 1 | 4 | 1 | 3 | 3 | 2 | 2 | 2 | CN | 4 | 4 | 3 | 3 | 3 | 2 | 2 | 2 |

## Comparison with current situation

The municipality, where the case study was carried out, placed seven-compartment containers for paper, plastic, glass, metal, vegetable oil, small electronic goods, and battery waste at 24 points within the district borders approximately three years ago. The municipality performance report stated that the average annual e-waste amount for recycling over the last three years was 500 kg [51]. If the difference between the model's values and the actual values is interpreted, it should first be noted that the model's values, like the total budget and container cost, are entirely hypothetical. At the same time, the capacity of the container is the main factor leading to the high e-waste values predicted by the model and expected for collection. The containers proposed in the study are only for the purpose of collecting e-waste, and their capacity is considered to be approximately 4.5 m³/1500 kg. It also focused on the collection of WEEE in the 2nd and 6th categories, which include medium-sized and voluminous e-waste such as televisions, monitors, and tablets. However, the containers currently used by the municipality consist of seven small compartments, and only one of these compartments (approximately 0.6 m³/150 kg) is reserved for e-waste. Although e-waste collection activities are partially carried out in the region covered by the study, they are inadequate, as can be understood by the annual average collected values.

The model suggested in the study looks from a different point and predicts how much e-waste could be collected if containers with the suggested size are set up at the locations chosen by the model.

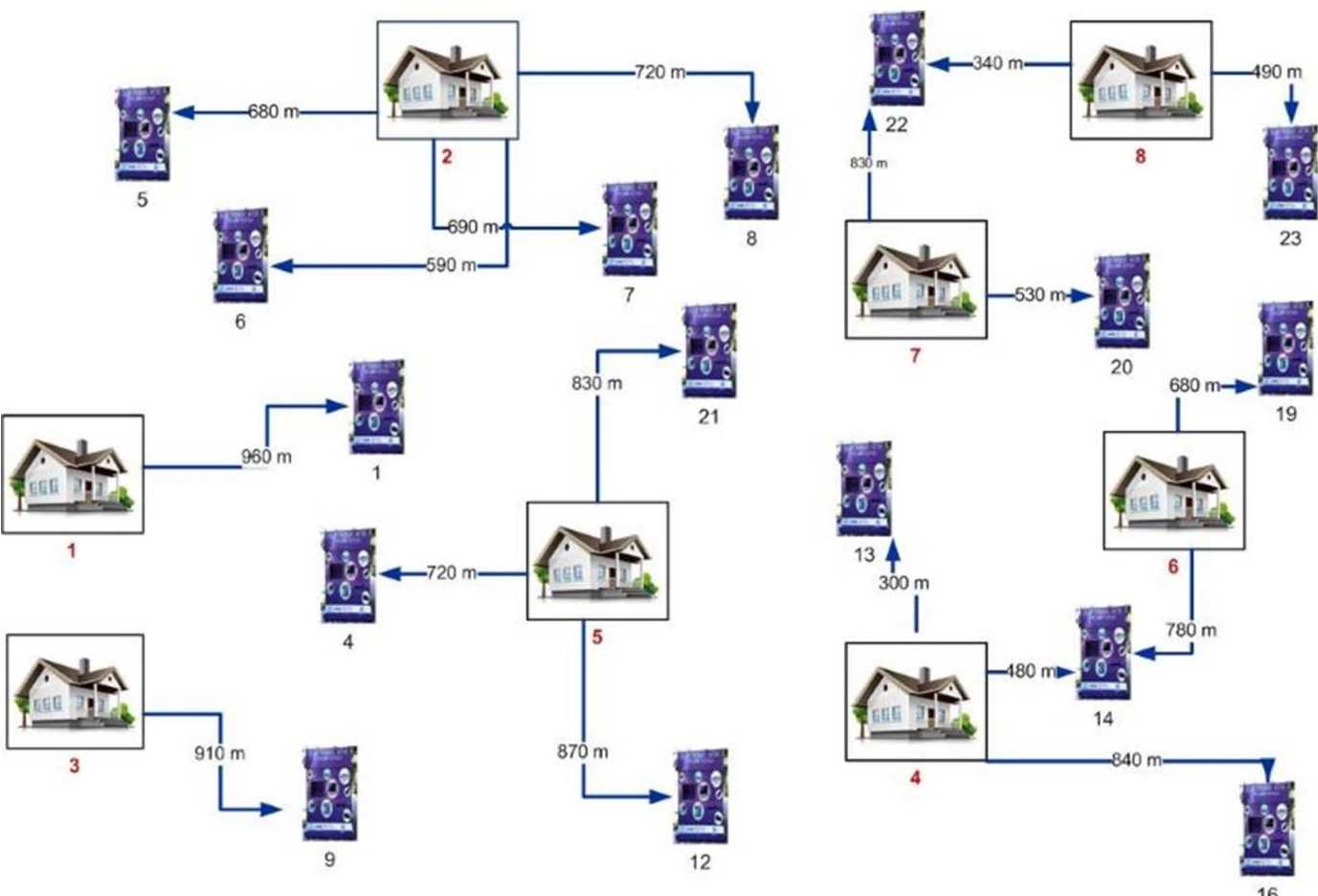

**Fig 5. Assignment results for maximum coverage distance of 1000 m (16 containers).**

## Discussion and conclusions

E-waste recycling is crucial to sustainability as it reduces waste, minimizes damage to the environment, conserves resources, and maintains a circular approach to electronics production vs. disposal.

As previously stated, many studies have demonstrated that e-waste recycling behaviors can be analyzed by accounting for a wide range of socio-economic and demographic factors, including income, gender, education level, age, environmental consciousness, habits, convenience, etc. The results reveal that the distinct characteristics of each country might influence individuals' participation in e-waste collection and recycling efforts in various manners.

Even though numerous studies have discussed these factors, none of them have directly included them in the waste container assignment model. The regional ratios derived from these factors have been integrated into the e-waste consumption capacity constraint through a harmonic mean. So, the proposed approach distinguishes itself from existing literature by incorporating factors influencing e-waste recycling into the e-waste container allocation model, addressing these factors at both macro and micro levels.

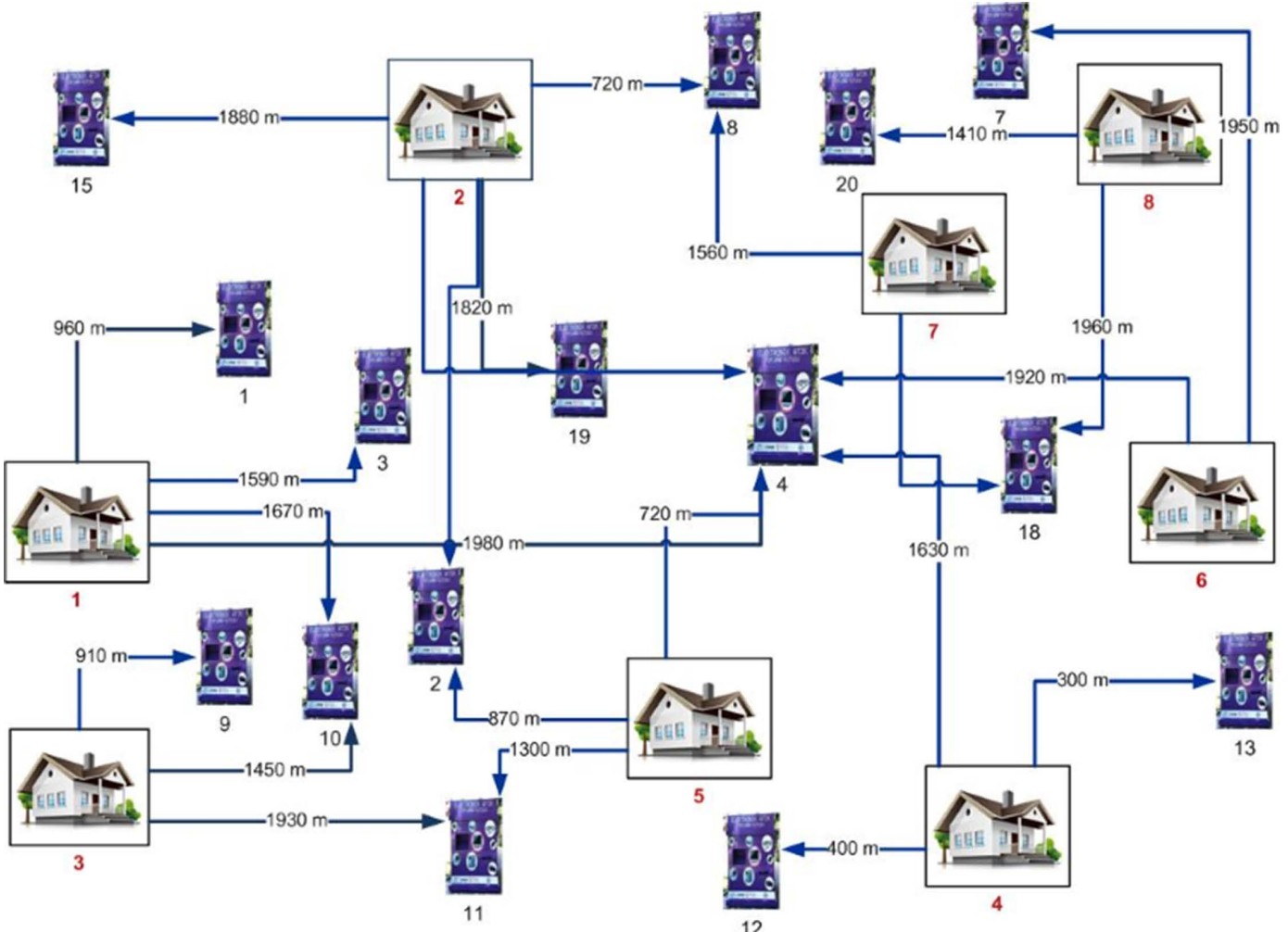

**Fig 6. Assignment results for maximum coverage distance of 2000 m (15 containers).**

The suggested approach varies from the conventional cluster coverage model since this model's objective function is to maximize the amount of e-waste. This use of maximum coverage distance is similar to the distributed energy network design study in [45].

Examining the theoretical framework, it becomes evident that while the model primarily addresses the assignment of waste containers, the findings suggest that the variables integrated into the proposed model have minimal influence on the quantity of containers deployed. The outcome aligns with prior expectations, given that the model incorporates the requirement of allocating a minimum of one container to each region. The variation in the number of containers appears to be primarily attributed to the adjustments made in the coverage distance incorporated within the model. The primary contribution of the proposed model lies in its ability to ascertain the quantity of waste that is expected to be collected which varies in regions with high and low e-waste production potential.

As mentioned before, waste collection management consists of many sequential steps, and each step has an impact on the next. The more accurately the amount of waste to be collected is estimated in the waste container placement process, which is considered the first step, the more accurately and effectively the policies regarding the next step, collection and separation/disposal processes, could be determined.

From a practical point of view, the proposed model's generic structure allows for its research findings to be applicable beyond a single region. This versatility enables policymakers in various local governments to incorporate both macro and micro variables, facilitating the development of first-step e-waste recycling policies. Ultimately, this approach aims to foster the establishment of more sustainable practices in e-waste recycling. Moreover, collaborative research efforts with local authorities are set to lead to more accurate outcomes.

In practice, it is seen that in terms of waste management, local governments often prefer intuitive and palliative solutions rather than systematic and permanent solutions. In general, local governments intuitively prefer to place waste recycling containers in crowded passageways like markets, parks, and ATMs. For instance, the municipality where the case study is carried out has placed seven-compartment containers for paper, plastic, glass, metal, vegetable oil, small electronic goods, and battery waste at 24 points within the district borders. However, separate containers focused solely on e-waste collection are not used. Moreover, since there is only one compartment allocated for e-waste, it only has a capacity to accommodate a very limited number of e-waste items, such as headphones, TV remote controls, speakers, mobile phones, or small computer parts. This situation also seriously affects the e-waste collection potential.

This study proposes a generic model that assigns the appropriate number of e-waste containers to the optimum points, considering the most appropriate coverage distance, according to the socio-demographic profile of local governments.

As suggestions for future research, factors affecting e-waste recycling would be included in the assignment model with different weights. Additionally, costs could be added to the model as a constraint, and the proposed approach could be used for different factors affecting recycling and for different categories of WEEE for different countries. Moreover, to evaluate and measure the success of recycling activities, the recycling rate, which calculates the percentage of waste, could be considered if the implementation of the proposed model could be carried out in a joint project with the local government. Since the demand generated at a node decreases exponentially with distance from the facility [52], a non-linear model might be more suitable for the assignment problem instead of using discrete values of Dc.

## Supporting information

**S1 Table. Number of houses for each neighborhood.**
(PDF)

**S2 Table. Distances between demand points and candidate container points.**
(PDF)

**S3 Table. Distribution of candidate containers by neighborhood.**
(PDF)

**S4 Table. Population for the selected neighborhoods and the amount of income per capita for each one.**
(PDF)

**S5 Table. Scoring for the education level.**
(PDF)

**S6 Table. Education level and age distribution by neighborhood.**
(PDF)

**S7 Table. Population, income, education, and age ratios of neighborhoods.**
(PDF)

**S8 Table. Ranking of regions according to harmonic mean values.**
(PDF)

## Author contributions

**Conceptualization:** Tuba Bazancir, Yeşim Ok.

**Data curation:** Tuba Bazancir.

**Formal analysis:** Yeşim Ok.

**Investigation:** Tuba Bazancir.

**Methodology:** Yeşim Ok.

**Software:** Yeşim Ok.

**Supervision:** Yeşim Ok.

**Validation:** Yeşim Ok.

**Visualization:** Tuba Bazancir, Yeşim Ok.

**Writing – original draft:** Yeşim Ok.

**Writing – review & editing:** Yeşim Ok.

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
