## [Decision Letter · Decision Letter 0]

20 Dec 2024

PONE-D-24-46286A case study of transferring the effect of demographic factors on e-waste recycling to the waste container assignment modelPLOS ONE

Dear Dr. OK,

Thank you for submitting your manuscript to PLOS ONE. After careful consideration, we feel that it has merit but does not fully meet PLOS ONE’s publication criteria as it currently stands. Therefore, we invite you to submit a revised version of the manuscript that addresses the points raised during the review process.

Please find the reviewers comments below and respond with a tracked changes file of your manuscript and point-by-point response to their concerns. You do not need to respond to reviewer 2 and 3 comments, as we have concerns about the potential of an AI tool was used to generate the review. We also note that reviewer 1 requests several citations are included. As always, we recommend that you please review and evaluate the requested works to determine whether they are relevant and should be cited. It is not a requirement to cite these works.

We look forward to receiving your revised manuscript.

Kind regards,

Joanna Tindall, PhD

Staff Editor

PLOS ONE

Journal Requirements:

2. Please ensure that you refer to Figure 2 in your text as, if accepted, production will need this reference to link the reader to the figure.

3. We note that Figures 4 in your submission contain [map/satellite] images which may be copyrighted. All PLOS content is published under the Creative Commons Attribution License (CC BY 4.0), which means that the manuscript, images, and Supporting Information files will be freely available online, and any third party is permitted to access, download, copy, distribute, and use these materials in any way, even commercially, with proper attribution. For these reasons, we cannot publish previously copyrighted maps or satellite images created using proprietary data, such as Google software (Google Maps, Street View, and Earth). For more information, see our copyright guidelines: http://journals.plos.org/plosone/s/licenses-and-copyright.

a. You may seek permission from the original copyright holder of Figures 4 to publish the content specifically under the CC BY 4.0 license.  

Reviewers' comments:

Reviewer's Responses to Questions

**Comments to the Author**

1. Is the manuscript technically sound, and do the data support the conclusions?

Reviewer #1: Yes

Reviewer #2: Yes

Reviewer #3: Yes

Reviewer #4: Yes

2. Has the statistical analysis been performed appropriately and rigorously? 

Reviewer #1: Yes

Reviewer #2: No

Reviewer #3: Yes

Reviewer #4: Yes

3. Have the authors made all data underlying the findings in their manuscript fully available?

Reviewer #1: No

Reviewer #2: Yes

Reviewer #3: Yes

Reviewer #4: Yes

4. Is the manuscript presented in an intelligible fashion and written in standard English?

Reviewer #1: Yes

Reviewer #2: No

Reviewer #3: Yes

Reviewer #4: Yes

5. Review Comments to the Author

Reviewer #1: • The abstract is comprehensive but could be more concise. Ensure it highlights key findings and the significance of the study more clearly.

• Enhance the discussion on the gap in the literature that this study aims to fill.

• The introduction provides a thorough background but lacks a clear statement of the research gap. Clarify the specific limitations of the existing models and how this paper aims to address these limitations. Including a direct statement of the research gap can strengthen the narrative.

• The literature review is quite comprehensive, but it could be enriched by adding more recent studies or cases from other countries.

• Authors are requested to add suggested published articles to improve the quality of the work, these are

1. E-waste circularity in India: identifying and overcoming key barriers, Journal of Material Cycles and Waste Management 26 (6), 3928-3945, https://doi.org/10.1007/s10163-024-02050-1.

2. Inconsistencies of e-waste management in developing nations - Facts and plausible solutions, Journal of Environmental Management, 261, 2020, 110234, https://doi.org/10.1016/j.jenvman.2020.110234.

3. Prioritizing Factors Affecting E-Waste Recycling in India: A Framework for Achieving a Circular Economy, Circular Economy and Sustainability, 1-21, https://doi.org/10.1007/s43615-024-00423-0.

• Compare the effectiveness of traditional and green methods with specific case studies or experimental data.

• Add research methodology flow chart of the study.

• Highlight the potential challenges in implementing the proposed model in real-world scenarios, such as data availability or political constraints.

• Write theoretical and practical implications of the work.

• The conclusion could be more concise. Focus on the key findings and the specific improvements your method offers over existing models. Refrain from repeating details already covered in previous sections.

Reviewer #2: Review of the Manuscript:

Technical Soundness

The study addresses an acknowledged gap in other related literatures where demographic factors have largely been excluded from container assignment models. The manuscript is technically sound in its formulation and presentation. The data presented align with the study's objectives. However, the lack of real-world validation (e.g., through collaboration with local governments) weakens the strength of the conclusions.

Experimental Rigor

I.

Controls and Comparisons: The study contrasts models with and without demographic factors to demonstrate the added value of their inclusion. This comparative approach acts as a form of control. For replication, while scenario analysis offers an implicit form of sensitivity analysis, the study lacks explicit replication or testing in varied real-world settings to confirm its robustness.

II.

Sample Sizes: The dataset (e.g., number of neighborhoods, container placement points) appears adequate for a simulation-based study. However, it does not detail whether the sample size of demographic inputs is statistically significant, which could influence generalizability.

III.

Limitations: The manuscript acknowledges limitations, such as the absence of real-world implementation and the need to test different weightings of factors. These acknowledgments strengthen the credibility of the conclusions.

Recommendations for Improvement

I.

This research should incorporate field tests or real-world validation to substantiate the model’s practical applicability.

II.

Include a discussion of how different weightings of demographic factors affect outcomes would provide greater insight into the model’s sensitivity.

III.

Incorporate cost as a constraint or objective to enhance the model's real-world relevance, as cost considerations are central to municipal decisions.

IV.

Involving policymakers or local governments in designing and validating the model could address the noted limitations and bridge the gap between theory and practice.

The study would benefit from additional real-world validation and expanded analysis to strengthen its practical implications. The conclusions are appropriately drawn from the data, but their broader applicability remains speculative without further empirical validation.

2. Has the statistical analysis been performed appropriately and rigorously?

The manuscript employs a mathematical model with a focus on optimization rather than traditional statistical hypothesis testing.

Areas Lacking Rigor and suggested corrections

i.

No Formal Statistical Tests:

The manuscript should include statistical tests (e.g., regression analysis, ANOVA, or chi-square tests) to validate relationships between demographic factors and e-waste generation or container assignment efficiency This could verify whether the variables selected significantly impact the outcomes. Include hypothesis testing to determine whether the inclusion of demographic factors significantly improves model performance.

ii.

Sample Size and Data Distribution:

There should be explicit discussion of whether the sample size (number of neighborhoods, regions, or households) is adequate to ensure robust modeling.

The distribution of demographic variables is not analyzed (e.g., whether they follow a normal distribution or have skewness). This omission could limit the validity of the harmonic mean approach.

Perform descriptive statistics to assess the distribution, variance, and potential outliers in the demographic variables. This would ensure the appropriateness of using the harmonic mean.

iii.

Sensitivity Analysis:

While scenario analysis is performed, there is no detailed sensitivity analysis of how much each factor (e.g., income, education, or age) influences the outcomes. Weighting different factors could reveal their relative importance and validate the robustness of the harmonic mean approach.

iv.

Correlation or Causation:

The manuscript does not statistically examine whether demographic factors are directly correlated with e-waste generation or container use. This assumption is critical for validating the inclusion of these variables in the model.

v.

Confidence Intervals or Error Margins:

The results of the model are presented as deterministic values (e.g., e-waste amounts or container placements) without confidence intervals or error margins. Adding these would improve the reliability and interpretability of the findings.

Analyze how changes in individual demographic factors or their weights affect the model's outcomes. This would validate the robustness of the results and reveal the most influential factors.

Include error margins or confidence intervals for predicted e-waste amounts or container placements to communicate the uncertainty in the model's predictions.

Incorporating these elements would strengthen the credibility and applicability of the findings.

3.

Is the manuscript presented in an intelligible fashion and written in standard English? Is the language used clear, correct, and unambiguous? Have all typographical or grammatical errors should be corrected?

Language and Presentation Review

The manuscript is generally well-organized and written in standard academic English, with a clear structure and logical flow of ideas. However, there are areas where clarity, conciseness, and grammatical accuracy could be improved. Below is an evaluation of its presentation:

Areas for Improvement

1.

Typographical and Grammatical Errors: Use Grammarly to identify and fix grammatical and typographical issues.

Examples of grammatical issues:

▪

"As the harmonic mean eliminates the extreme points and reduces the effect of averaging the high value between the data, it is preferred in this study for factors combination."

Suggested revision: "The harmonic mean is preferred in this study as it eliminates extreme values and reduces the effect of averaging high data points."

▪

"Local governments could utilize these findings as a reference to create more feasible policies for the following steps of e-waste recycling."

▪

Suggested revision: "Local governments could use these findings as a reference to develop more effective e-waste recycling policies."

Several sentences are verbose or repetitive and can be streamlined for conciseness.

Inconsistent Word Choice:

Phrases such as "to gain insight while determining" and "considerably more realistic results" could be replaced with more precise expressions like "to inform decisions" or "more accurate results."

Some phrases, like "not proven to be accurate in a joint project," are awkwardly phrased. Suggested revision: "The findings have not been validated through collaboration with local authorities."

Ambiguity:

Some technical explanations, while accurate, are overly complex:

▪

"The model comes to the fore with a different perspective proposal as it creates an e-waste demand based on the different characteristics of each neighborhood."

▪

Suggested revision: "The model provides a novel perspective by estimating e-waste demand based on neighborhood-specific characteristics."

Redundancy:

Certain ideas are repeated, such as the importance of demographic factors or the generic applicability of the model. These repetitions could be condensed to improve readability.

Typographical Errors:

Occasional minor errors, such as missing articles ("the" or "a") and punctuation issues, need correction. For example:

▪

"with every passing day" should be "with each passing day."

▪

Missing commas in long sentences can make them harder to read.

Abstract and Conclusions:

The abstract is informative but overly dense. Reorganize the abstract to present the problem, methodology, results, and implications in a succinct, easily digestible format.

Simplifying the language while retaining key points. The conclusion could be refined to avoid repetition of previously stated points, focusing instead on actionable insights and future research directions.

Related Works

Add these recent research papers related to e-waste recycling, emphasizing demographic factors, consumer behavior, and optimization techniques to your study:

1. "Exploring Factors of E-Waste Recycling Intention: The Case of Generation Y"

Authors: Multiple authors Published: 2023 (PLOS ONE) Relevance: Provides insights into generational differences in recycling behavior, which can inform targeted policies for improving e-waste collection strategies. Link: PLOS ONE Article

2. "The Determinants of Consumers' E-Waste Recycling Behavior through the Lens of Extended Theory of Planned Behavior"

Authors: Nur Shafeera Mohamad et al. Published: 2022 (Sustainability) Relevance: Highlights the importance of convenience and moral responsibility in recycling, which are critical for designing practical e-waste recycling systems. Link: MDPI Article

3. "Determinants of Individuals' E-Waste Recycling Decision: A Case Study from Romania"

Authors: Corina Ioanăș et al. Published: 2020 (Sustainability) Abstract: This paper investigates how demographic variables, such as age and gender, influence e-waste recycling behaviors in Romania. It also evaluates the role of social media and governmental actions in shaping recycling decisions. Relevance: Emphasizes the interplay of socio-demographic factors and policy measures in recycling, which can be integrated into container assignment models. Link: MDPI Article

4. "Prioritizing Factors Affecting E-Waste Recycling in India: A Framework for Decision-Making"

Authors: Multiple authors Published: 2021 (Springer) Relevance: Offers valuable insights into context-specific challenges and solutions for e-waste recycling in developing economies. Link: Springer Article

5. "Driving Factors of E-Waste Recycling Rates in 30 European Countries: A Quantitative Analysis"

Authors: Various Published: 2021 (Springer) Relevance: Offers a macro-level perspective on e-waste recycling, which complements localized studies like the one in the manuscript. Link: Springer Article

The reviewed papers collectively emphasize the importance of: Demographic Factors, Behavioral Theories, Policy Implications and Cultural and Regional Differences.

Recommendations: The manuscript is intelligible, clear, and written in standard English but requires minor revisions for grammatical correctness, conciseness, and clarity. Addressing the issues highlighted above will significantly improve its readability and presentation quality, making it more polished for publication.

Reviewer #3: 1. Is the manuscript technically sound, and do the data support the conclusions?

The manuscript is technically sound in terms of its mathematical modeling approach and the incorporation of demographic factors into e-waste container assignment models. The use of harmonic means to account for socio-demographic variations is innovative and addresses a gap in the literature. The case study effectively demonstrates the application of the model, and the results support the claim that demographic factors improve the realism of e-waste collection estimations. However, the study acknowledges a limitation: the results have not been validated in collaboration with local governments. This reduces the practical applicability of the findings, which should be highlighted.

2. Has the statistical analysis been performed appropriately and rigorously?

The analysis includes a robust formulation of constraints and optimization objectives, and the harmonic mean calculation is methodologically sound. The study explores scenarios with varying coverage distances to test the model’s adaptability. However, the lack of real-world validation or sensitivity analysis on the weights assigned to demographic factors limits the generalizability of conclusions. Future studies might benefit from including statistical tests or validations to confirm model predictions.

3. Have the authors made all data underlying the findings in their manuscript fully available?

The authors state that all relevant data are available within the manuscript and its supporting information. Parameters like demographic data, e-waste generation rates, and container capacities are clearly described and included in the appendices. This transparency is commendable and aligns with open data principles. No concerns arise regarding data availability.

4. Is the manuscript presented in an intelligible fashion and written in standard English?

The manuscript is presented in a clear and comprehensible manner. The technical terms are adequately explained, making the work accessible to a broad audience. There are minor grammatical and stylistic errors, but they do not impede understanding. For example, some sentences could be rephrased for conciseness and clarity. A thorough editorial review would enhance the readability further.

Recommendations:

• Validation: Collaborate with local governments for real-world application to strengthen the manuscript's practical contributions.

• Sensitivity Analysis: Explore the impact of varying weights assigned to socio-demographic factors on the model’s outcomes.

COMMENT: Address minor language issues for improved clarity and readability.

The manuscript is overall a significant contribution to the field of e-waste management, combining socio-demographic insights with mathematical modeling.

Reviewer #4: Dear Authors,

Your submission, "A Case Study of Transferring the Effect of Demographic Factors on E-Waste Recycling to the Waste Container Assignment Model," has been received and reviewed. An important and pertinent topic in waste management research is the study's innovative method of incorporating demographic variables into e-waste container assignment models. My detailed comments and suggestions are in the attached file.

6. PLOS authors have the option to publish the peer review history of their article (what does this mean? ). If published, this will include your full peer review and any attached files.

**Do you want your identity to be public for this peer review?** For information about this choice, including consent withdrawal, please see our Privacy Policy .

Reviewer #1: No

Reviewer #2: **Yes: ** Prof. Susan Konyeha

Reviewer #3: **Yes: ** OISE GODFREY PERFECTSON

Reviewer #4: No

---

## [Author Response · Author response to Decision Letter 1]

15 Jan 2025

Response to Editor

Comment 1:

Response 1:

PLOS ONE's style requirements have been checked, and necessary adjustments have been made.

Comment 2:

Please ensure that you refer to Figure 2 in your text as, if accepted, production will need this reference to link the reader to the figure.

Response 2:

Figure 2 is linked to the body text of the manuscript previous paragraph on page 4.

Comment 3:

We note that Figures 4 in your submission contain [map/satellite] images which may be copyrighted.

We require you to either (1) present written permission from the copyright holder to publish these figures specifically under the CC BY 4.0 license, or (2) remove the figures from your submission.

Response 3:

The Google Earth screenshot in Figure 4 does not serve as a primary figure for the study; it has been included solely to offer a basic illustration. In this context, as you have indicated, it was extracted from the main text and substituted with a visual representation illustrating the relationship between candidate container points and demand points.

Figure 4. Candidate container and demand points relationship illustration

Comment 4

Please include captions for your Supporting Information files at the end of your manuscript, and update any in-text citations to match accordingly as stated in https://journals.plos.org/plosone/s/supporting-information

Response 4

We have included captions for Supporting Tables at the end of our manuscript, and updated all in-text citations to match accordingly.

Comment 5:

Response 5:

The reference list has been checked. No retracted article is found.

The following references have been added to the manuscript in response to referee comments:

1. Pakpour AH, Zeidi IM, Emamjomeh MM, Asefzadeh S, Pearson H. Household waste behaviours among a community sample in Iran: An application of the theory of planned behaviour. Waste Manag. 2014 Jun 1;34(6):980–6.

2. Sidique SF, Lupi F, Joshi S V. The effects of behavior and attitudes on drop-off recycling activities. Resour Conserv Recycl. 2010 Jan 1;54(3):163–70.

3. Xu L, Ling M, Lu Y, Shen M. Understanding Household Waste Separation Behaviour: Testing the Roles of Moral, Past Experience, and Perceived Policy Effectiveness within the Theory of Planned Behaviour. Sustainability. 2017 Apr 17;9(4):625.

4. Quynh Vo C, Samuelsen PJ, Leikny Sommerseth H, Wisløff T, Wilsgaard T, Elise Eggen A. Comparing the sociodemographic characteristics of participants and non-participants in the population-based Tromsø Study. BMC Public Health. 2023 Dec 1;23(1).

Responses to Reviewers

Reviewer #1:

Comment 1:

The abstract is comprehensive but could be more concise. Ensure it highlights key findings and the significance of the study more clearly.

Response 1:

This paragraph is added to the Abstract.

“The main contribution of the proposed model is that significantly more realistic results regarding the amount of e-waste to be collected can be obtained when factors indicating regional differences are included in the container assignment model. Thus, local governments could utilize these findings as a reference to create more sustainable policies for the next steps of e-waste recycling. The model application is demonstrated through a case study for a local government in Turkey.”

Comment 2:

Enhance the discussion on the gap in the literature that this study aims to fill.

Response 2:

This paragraph is added to the last paragraph of Literature:

Previous studies examining the factors influencing e-waste recycling behavior are predominantly survey-based, aimed at delineating the consumer profile. In summary, despite numerous studies addressing the determinant factors, none have concentrated on the impact of these factors on recycling behavior. Essentially, the main motivation of this study is to create a general framework that would aid in the development of local e-waste collection policies based on the basic socio-demographic factors that cause variations from region to region in the recycling rate of e-waste.

Comment 3:

The introduction provides a through background but lacks a clear statement of the research gap. Clarify the specific limitations of the existing models and how this paper aims to address these limitations. Including a direct statement of the research gap can strengthen the narrative.

Response 3:

This paragraph is added to the Introduction to address the gap.

While survey-based studies have identified the qualitative and quantitative factors influencing e-waste recycling behavior, a significant gap remains in establishing a framework that incorporates these factors into the planning of the e-waste recycling process. This study seeks to address this gap, specifically by incorporating quantifiable socio-demographic factors into the waste container allocation model, addressing these factors at both macro and micro levels.

Comment 4:

The literature review is quite comprehensive, but it could be enriched by adding more recent studies or cases from other countries.

Response 4:

The following studies are added to enrich the literature.

1. Pakpour AH, Zeidi IM, Emamjomeh MM, Asefzadeh S, Pearson H. Household waste behaviours among a community sample in Iran: An application of the theory of planned behaviour. Waste Manag. 2014 Jun 1;34(6):980–6.

2. Sidique SF, Lupi F, Joshi S V. The effects of behavior and attitudes on drop-off recycling activities. Resour Conserv Recycl. 2010 Jan 1;54(3):163–70.

3. Xu L, Ling M, Lu Y, Shen M. Understanding Household Waste Separation Behaviour: Testing the Roles of Moral, Past Experience, and Perceived Policy Effectiveness within the Theory of Planned Behaviour. Sustainability. 2017 Apr 17;9(4):625.

4. Quynh Vo C, Samuelsen PJ, Leikny Sommerseth H, Wisløff T, Wilsgaard T, Elise Eggen A. Comparing the sociodemographic characteristics of participants and non-participants in the population-based Tromsø Study. BMC Public Health. 2023 Dec 1;23(1).

Comment 5:

Authors are requested to add suggested published articles to improve the quality of the work, these are

1. E-waste circularity in India: identifying and overcoming key barriers, Journal of Material Cycles and Waste Management 26 (6), 3928-3945, https://doi.org/10.1007/s10163-024-02050-1.

2. Inconsistencies of e-waste management in developing nations - Facts and plausible solutions, Journal of Environmental Management, 261, 2020, 110234, https://doi.org/10.1016/j.jenvman.2020.110234.

3. Prioritizing Factors Affecting E-Waste Recycling in India: A Framework for Achieving a Circular Economy, Circular Economy and Sustainability, 1-21, https://doi.org/10.1007/s43615-024-00423-0.

Response 5:

The suggested studies have been examined.

In the study titled "Inconsistencies of e-waste management in developing nations: Facts and plausible solutions," ten major deficiencies in e-waste management at the strategic level have been identified. In this context, our study does not overlap in terms of scope since it focuses on the factors affecting e-waste recycling as the first-step policies.

In the study titled "E-waste circularity in India: identifying and overcoming key barriers," obstacles such as lack of incentives for e-waste recycling in India, customer awareness, and government policies were addressed. Also, in the study titled "Prioritizing Factors Affecting E-Waste Recycling in India: A Framework for Achieving a Circular Economy," from a reverse perspective, eleven critical factors affecting e-waste recycling management in India were identified, emphasizing circular economy adoption incentives, government policies, and customer awareness.

Since both of these two studies do not overlap in terms of scope with our study, it was not deemed necessary to add them to the literature.

Comment 6:

Compare the effectiveness of traditional and green methods with specific case studies or experimental data.

Response 6:

First, the approach we propose is preferred not to be called a completely "green method" but rather a "sustainable method."

In addition, the biggest difference with the traditional approach is related to the amount of e-waste expected to be collected. Namely, when the factors affecting recycling are not included in the container assignment model, the amount of waste to be obtained is considerably higher than the amount of waste to be obtained when they are included. This situation has already been mentioned in the results-scenario analysis section, and the relevant data has been given in Table 2.

It can be concluded that the approach, which is thought to be effective on the first-step e-waste recycling decisions of local decision-makers, can also prevent the assignment of extra containers from this perspective and thus prevent unnecessary costs.

This following statement is added to the paragraph, after Table 2.

“So, this proposed approach, deemed effective in the initial phase of e-waste recycling decisions by local authorities, can prevent the allocation of unnecessary containers and associated costs.”

Comment 7:

Add research methodology flow chart of the study.

Response 7:

The flow chart of the research methodology has already been added as

“Figure 3. The general flow of the research methodology.” The name is changed.

Comment 8:

Highlight the potential challenges in implementing the proposed model in real-world scenarios, such as data availability or political constraints.

Response 8:

Since the proposed model takes into account quantitative values related to e-waste recycling behavior and is not survey-based, accessing this data is relatively easier than relative factors such as environmental awareness, habits, convenience, etc. In this context, there are no restrictions in accessing and using the data. Regarding the model application, it has been emphasized before that working in a joint project with the local government would yield more realistic results and that this is a weakness of the study.

Comment 9:

Write theoretical and practical implications of the work.

Response 9:

The following paragraphs are added to the Conclusion Section for theoretical and practical implications.

Examining the theoretical framework, it becomes evident that while the model primarily addresses the assignment of waste containers, the findings suggest that the variables integrated into the proposed model have minimal influence on the quantity of containers deployed. The outcome aligns with prior expectations, given that the model incorporates the requirement of allocating a minimum of one container to each region. The variation in the number of containers appears to be primarily attributed to the adjustments made in the coverage distance incorporated within the model. The primary contribution of the proposed model lies in its ability to ascertain the quantity of waste that is expected to be collected which varies in regions with high and low e-waste production potential.

From a practical point of view, the proposed model's generic structure allows for its research findings to be applicable beyond a single region. This versatility enables policymakers in various local governments to incorporate both macro and micro variables, facilitating the development of first-step e-waste recycling policies. Ultimately, this approach aims to foster the establishment of more sustainable practices in e-waste recycling. Moreover, collaborative research efforts with local authorities are set to lead to more accurate outcomes.

Comment 10:

The conclusion could be more concise. Focus on the key findings and the specific improvements your method offers over existing models. Refrain from repeating details already covered in previous sections.

Response 10:

Some paragraphs marked in the revised manuscript in the conclusion section were removed to simplify the text. In addition, the paragraphs in Comment 9 and the sections requested by the other referee were added, and the Discussion and Conclusion sections were rearranged.

Reviewer #2 and Reviewer #3:

The Editor indicated that since they are concerned about the possibility that an AI tool was used to create the review, I do not need to reply to the remarks made by reviewers 2 and 3.

Reviewer #4:

Dear Authors,

Your submission, "A Case Study of Transferring the Effect of Demographic Factors on E-Waste Recycling to the Waste Container Assignment Model," has been received and reviewed. An important and pertinent topic in waste management research is the study's innovative method of incorporating demographic variables into e-waste container assignment models.

My detailed comments and suggestions are in the attached file.

Comment 1:

This statement “Among these factors, it is obvious that the income and educational levels of electronic consumers, as well as the age and population size distribution in the surrounding area, are all significant. “ seems to be the author's opinion, so it would be better if the author could refer to the most relevant references.

Response 1:

In place of the abovementioned phrase, the following phrase is added to the manuscript:

A significant portion of the studies on recycling behavior state that socio-demographic characteristics have a significant effect on recycling behavior(1–4), while others claim that these factors play only a minimal role (5).

Comment 2:

It would be better if the author could explain what "sociodemographic characteristics" are and so on first.

Response 2:

An explanation for sociodemographic characteristics has been added at the beginning of the paragraph with a reference (6).

Socio-demographic characteristics encompass a range of social and demographic factors, which are often measured by an individual's age, level of education, occupation, and income.

Comment 3:

This sentence is more appropriate in the introduction.

The main purpose of this study is to create a generic assignment model based on the factors affecting electronic waste generation.

Response 3:

The relevant sentence was removed from the materials and methods section and combined with a similar sentence in the introduction.

Comment 4:

Is figure 2 not linked to the body text of the manuscript?

Figure 2. Country-based differences affecting WEEE consumption in the macro sense and Socio-economic/demographic factors affecting WEEE behavior in the micro sense

Response 4:

Figure 2 is linked to the body text of the manuscript previous paragraph on page 4.

Comment 5:

Please, Could you kindly check that "factors affecting e-waste behavior" should appear in step 1 of figure 3?

Figure 3. The general flow of the proposed approach

Response 5:

Figure 3 has been revised by removing the statement "factors affecting e-waste behavior" in step 1.

Comment 6:

So, can you explain in more detail whether this research is only for e-waste from the televisions/monitors and informatics-telecommunication equipment category?

Response 6:

The following paragraph was added to explain this research is only for e-waste from the televisions/monitors and informatics-telecommunication equipment category.

“E-waste encompasses an extensive variety of products. As previously stated, e-waste is categorized into six classes in accordance with the European Union WEEE Directive (Fig. 1). This study focuses on the collection of e-waste categories ranked second (displays) and sixth (telecommunication equipment). The application is restricted to these two categories because items like monitors, tablets, mobile phones, and headphones are of sizes and weights portable by household and the container capacities designated in the model are appropriate for this context. For instance, items like refrigerators in category 1 and washing machines in category 4 are not suitable for households to transport to the nearest disposal container. Once more, light b

---

## [Decision Letter · Decision Letter 1]

24 Apr 2025

PONE-D-24-46286R1

A case study of transferring the effect of demographic factors on e-waste recycling to the waste container assignment model

PLOS ONE

Dear Dr. OK,

Thank you for submitting your manuscript to PLOS ONE. After careful consideration, we feel that it has merit but does not fully meet PLOS ONE’s publication criteria as it currently stands. Therefore, we invite you to submit a revised version of the manuscript that addresses the points raised during the review process.

We look forward to receiving your revised manuscript.

Kind regards,

Mwazvita TB Dalu, PhD

Academic Editor

PLOS ONE

Journal Requirements:

Additional Editor Comments:

Comments from the editorial office: Upon internal evaluation of the reviews provided, we kindly request you to disregard the reviewer report provided by Reviewer 1. No amendments are required in response to reviewer 3’s comments’

Thank you for your submission. Your manuscript has now been reviewed and come in with a minor revision. Please address the provided comments from your reviewers and resubmit your manuscript.

Reviewers' comments:

Reviewer's Responses to Questions

**Comments to the Author**

1. If the authors have adequately addressed your comments raised in a previous round of review and you feel that this manuscript is now acceptable for publication, you may indicate that here to bypass the “Comments to the Author” section, enter your conflict of interest statement in the “Confidential to Editor” section, and submit your "Accept" recommendation.

Reviewer #3: All comments have been addressed

Reviewer #4: All comments have been addressed

2. Is the manuscript technically sound, and do the data support the conclusions?

Reviewer #3: Yes

Reviewer #4: Yes

3. Has the statistical analysis been performed appropriately and rigorously? 

Reviewer #3: Yes

Reviewer #4: (No Response)

4. Have the authors made all data underlying the findings in their manuscript fully available?

Reviewer #3: Yes

Reviewer #4: (No Response)

5. Is the manuscript presented in an intelligible fashion and written in standard English?

Reviewer #3: Yes

Reviewer #4: No

6. Review Comments to the Author

Reviewer #3: (No Response)

Reviewer #4: The manuscript has been revised based on the previous reviewers' comments. However, further improvements are needed to clarify the research objectives, contributions, and consistency of terminology (diction) to make it easier for readers to understand. For example, what definitions do the terms "micro sense" and "macro sense" refer to? Additionally, the authors need to recheck the writing standards and grammar.

7. PLOS authors have the option to publish the peer review history of their article (what does this mean? ). If published, this will include your full peer review and any attached files.

**Do you want your identity to be public for this peer review?** For information about this choice, including consent withdrawal, please see our Privacy Policy .

Reviewer #3: **Yes: ** Godfrey Perfectson Oise

Reviewer #4: No

---

## [Author Response · Author response to Decision Letter 2]

13 Jun 2025

As mentioned in the "Additional Editor Comments," we ignored the reports from Reviewers 1 and 3. We solely based our considerations and responses on the editor's reviewer report.

We tried to produce the most appropriate solutions to your comments by responding to all the questions step by step.

As a note, Figures and Additional Tables are as they have been uploaded before; no changes have been made to them.

---

## [Editor Report · Decision Letter 2]

30 Jul 2025

A case study of transferring the effect of demographic factors on e-waste recycling to the waste container assignment model

PONE-D-24-46286R2

Dear Dr. OK,

We’re pleased to inform you that your manuscript has been judged scientifically suitable for publication and will be formally accepted for publication once it meets all outstanding technical requirements.

Kind regards,

Mwazvita TB Dalu, PhD

Academic Editor

PLOS ONE
---

## [Editor Report · Acceptance letter]

PONE-D-24-46286R2

PLOS ONE

Dear Dr. Ok,

I'm pleased to inform you that your manuscript has been deemed suitable for publication in PLOS ONE. Congratulations! Your manuscript is now being handed over to our production team.

Kind regards,

on behalf of

Dr. Mwazvita TB Dalu

Academic Editor

PLOS ONE